# Co-Encapsulation and Co-Delivery of Peptide Drugs via Polymeric Nanoparticles

**DOI:** 10.3390/polym11020288

**Published:** 2019-02-08

**Authors:** Ma Rie Kim, Teng Feng, Qian Zhang, Ho Yin Edwin Chan, Ying Chau

**Affiliations:** 1Department of Chemical and Biological Engineering, The Hong Kong University of Science and Technology, Clear Water Bay, Hong Kong, China; mrkim@connect.ust.hk (M.R.K.); tfeng@connect.ust.hk (T.F.); 2School of Life Sciences, Faculty of Science, The Chinese University of Hong Kong, Shatin, N.T., Hong Kong, China; couragec@163.com (Q.Z.); hyechan@cuhk.edu.hk (H.Y.E.C.); 3Gerald Choa Neuroscience Centre, The Chinese University of Hong Kong, Shatin, N.T., Hong Kong, China

**Keywords:** co-encapsulation, co-delivery, nanoparticles, peptide delivery, polymeric drug delivery system, double emulsion-solvent evaporation technique(s)

## Abstract

Combination therapy is a promising form of treatment. In particular, co-treatment of P3 and QBP1 has been shown to enhance therapeutic effect in vivo in treating polyglutamine diseases. These peptide drugs, however, face challenges in clinical administration due to poor stability, inability to reach intracellular targets, and lack of method to co-deliver both drugs. Here we demonstrate two methods of co-encapsulating the peptide drugs via polymer poly(ethylene glycol)-block-polycaprolactone (PEG-b-PCL) based nanoparticles. Nanoparticles made by double emulsion were 100–200 nm in diameter, with drug encapsulation efficiency of around 30%. Nanoparticles made by nanoprecipitation with lipid 1-palmitoyl-2-oleoyl-sn-glycero-3-phospho-(1′-rac-glycerol) (POPG) were around 250–300 nm in diameter, with encapsulation efficiency of 85–100%. Particles made with both formulations showed cellular uptake when decorated with a mixture of peptide ligands that facilitate endocytosis. In vitro assay showed that nanoparticles could deliver bioactive peptides and encapsulation by double emulsion were found to be more effective in rescuing cells from polyglutamine-induced toxicity.

## 1. Introduction

Combination drug delivery systems have risen to interest as cocktail chemotherapy started to be explored for enhanced treatment [1,2]. Although combination therapy has been of great interest in cancer therapy, other potential applications exist. In particular, two peptide drugs for polyglutamine disease treatment have shown potential for synergistic effect after co-administration [3]. Peptide therapeutics P3 [3] and QBP1 [4,5] have been developed to suppress—in the nuclei and the cytoplasm, respectively—the toxicity of expanded cytosine-adenine-guanine (CAG) repeats of polyglutamine diseases. Co-treatment of P3 and QBP1 was previously demonstrated by us to significantly rescue polyglutamine toxicity compared to single administration of either peptide [3]. These peptide drugs, however, face challenge in clinical administration due to their susceptibility to enzymatic degradation [6] and lack of ability to penetrate cell membrane for cellular uptake. 

To compensate for this drawback, polymeric drug delivery systems can be utilised. Polymers are extensively studied as drug delivery systems [7,8]. Polymers have large capacity for functionalization, which can be utilised to conjugate targeting ligands to improve biodistribution. Nanoparticles can be synthesised to co-encapsulate P3 and QBP1 into the same nanoparticle. Hence, if a nanoparticle is taken up by a cell, both peptides will be released in that cell. This aspect is a great advantage in studying the effects of co-administration of two drugs. In the previous study [3], the co-administration of P3 and QBP1 was performed by applying peptides using separate transfection agents. Due to this delivery method, not all cells can be assumed to have both peptides simultaneously, while co-encapsulated nanoparticles will simultaneously deliver drugs into the cell. Polymeric nanoparticles can also compensate the disadvantages of peptide therapeutics by increasing bioavailability, exhibiting controlled and sustained drug release, and enhancing cellular uptake [9]. We aimed to design nanocarriers that can co-encapsulate and co-deliver both drugs; have high drug encapsulation efficiency and defined drug release profile; exhibit low polydispersity, weak positive surface charge (±20 mV), and suitable diameter (100–200 nm) to avoid renal clearance and detection by reticuloendothelial system [10,11]. To lay ground for later investigation on brain delivery efficacy, we also functionalize the polymers with ligands for targeted drug delivery at both tissue and cellular levels. We would like to note that the focus of this paper was to develop polymeric carriers and to evaluate the associated particle preparation process. The goal here is to co-encapsulate two hydrophilic peptide drugs, preserve their activity, and demonstrate the ability to deliver them to the intracellular space for functions under in vitro settings. These polymeric nanoparticles will have potential application for brain delivery, but such evaluation will need to be performed in vivo, which is out of the scope of the current investigation. 

In this paper, we present two methods, double emulsion method and nanoprecipitation method with POPG (1-palmitoyl-2-oleoyl-sn-glycero-3-phospho-(1′-rac-glycerol)), to create nanoparticles that can co-encapsulate and release hydrophilic peptide drugs. Poly(ethylene glycol)-block-polycaprolactone (PEG-b-PCL) was used to synthesise polymeric nanoparticles as PEG and PCL have been reported to be biocompatible and PCL was biodegradable without acidic by-product [12,13]. PEG has been reported to prolong systemic circulation by creating a hydrophilic barrier around the particle and to stabilise nanoparticles by steric stabilisation [7]. The rationale behind incorporation of POPG into nanoprecipitation was to enhance the drug encapsulation efficiency (E.E.) by utilising ionic interaction between the negative charges of POPG and the positive charges of the drugs. For cell penetration, PEG-b-PCL were conjugated with a mixture of peptide ligands to present the tissue and cell-targeting moieties on the particle surface. We compared the drug encapsulation efficiency, drug release profile, nanoparticle cellular uptake, and bioactivity of the two methods in vitro. 

## 2. Materials & Methods

### 2.1. Materials

P3V8, a variant of P3 [3,14], was used for co-encapsulation with QBP1. P3V8 (acetyl-DGKSKGIAYIEFK-amide, 1496.7 g/mol, +1 net charge at pH 7, solubility in water 143 mg/mL) and QBP1 (SNWKWWPGIFD, 1435.6 g/mol, 0 net charge at pH 7, solubility in water 420 mg/mL) were purchased from GL Biochem Ltd (Shanghai, China) (>95% purity). TGN (TGNYKALHPHNGC-amide), COG133 (LRVRLASHLRKLRKRLLC-amide), and RXR [(RXRRBR)_2_XBC-amide] were synthesized in the Chau laboratory using microwave assisted solid-phase peptide synthesis [15]. Methoxy poly(ethylene glycol) (mPEG, $M_{w}$= 2000 Da), ε-caprolactone (97%), tin(II) 2-ethylhexanoate [Sn(Oct)2] (95%), 2,2-dimethoxy-2-phenylacetophenone (DMPA, 99%) were all purchased from Sigma-Aldrich (St. Louis, MO, USA). Allyl-PEG-OH ($M_{w}$ = 2,000 Da) was purchased from Shanghai Jinpan Biotech Inc. (Shanghai, China). 1-Palmitoyl-2-oleoyl-sn-glycero-3-phospho-(1′-rac-glycerol) (POPG) (sodium salt) was purchased from Avanti Polar Lipids (Alabaster, AL, USA). Amicon Ultra-2mL 100 kDa centrifugal filters were purchased from MilliporeSigma (Burlington, MA, USA).

### 2.2. Synthesis of mPEG-b-PCL, Allyl-PEG-b-PCL, TGN-PEG-b-PCL, COG133-PEG-b-PCL, and RXR-PEG-b-PCL

Polymer synthesis and conjugation of ligands were conducted based on an established protocol [16].

### 2.3. Preparation of NPs by Double Emulsion 

To create nanoparticles by double emulsion (DE NPs), peptides (3 mg P3V8 and 3 mg QBP1) were dissolved in 0.2 mL water (w1 phase) and polymers (1–10 mg) were dissolved in 1 mL organic solvent (o phase). The ratio of w1 phase to o phase was 1:5. After adding the w1 phase into the o phase, a probe sonicator was used to sonicate the mixture, making the first emulsion. This first emulsion was then added to the w2 phase (4 mL water) and sonicated. The ratio of o phase to w2 phase was 1:4. Organic solvent was removed from the second emulsion using rotary evaporator, yielding polymer-matrix nanoparticles suspended in water (Figure 1).

### 2.4. Preparation of NPs by Nanoprecipitation in the Presence of POPG

POPG NPs were prepared by two-step nanoprecipitation (Figure 2). Peptides (0.75 mg P3V8 and 0.75 mg QBP1) were dissolved in 1 mL water and was adjusted to the desired pH by spiking 1 M HCl or 1 M NaOH. POPG (2.73 mg) and polymer (1.25 mg) were dissolved in 0.1 mL tetrahydrofuran (THF) in separate oil phases. The ratio of oil phase to aqueous phase was 1:5. While vortexing the aqueous phase, the POPG oil phase was added. After vortexing for 30 s, the polymer oil phase was added and vortexed for 30 s. The suspension was stirred at room temperature for 1 hour to remove THF by evaporation.

### 2.5. Nanoparticle Characterization

Dynamic light scattering (DLS) was used to characterize nanoparticles (ZetaPlus, Brookhaven Instruments Corporation, Holtsville, NY, USA). Mean diameter by number, mean diameter by intensity, polydispersity index (PDI), and zeta potential were measured for each nanoparticle formulation. Additional characterizations of morphology and size by transmission electron microscopy (TEM) were performed (JEOL JEM-2010, JEOL, Tokyo, Japan). 

### 2.6. Drug Encapsulation Efficiency and Release Study

Nanoparticles were separated from non-encapsulated, free peptides by centrifugal filters. After centrifuging at 3600× *g* for 15 min, the filtrate of free peptide solution was retrieved. The filter units were flipped and centrifuged at 1900× *g* for 5 min to recover the particle concentrate. The peptide contents in the filtrates were determined by high-performance liquid chromatography (HPLC). Experiments were done to verify that peptides do not bind to the centrifugal filters and that nanoparticles can be effectively retrieved from filters. Briefly, peptide solution of known concentration was filtered through the centrifugal filters using the same centrifugation parameters. The filtrate was tested for peptide concentration using HPLC. The measured peptide concentration of the filtrate was same as the original concentration. The drug encapsulation efficiency (E.E.) was determined using the following equation:
$$E.E.\left(\%\right)=\frac{Mass\;of\;peptide\;in\;initial\;formulation-Mass\;of\;free\;peptide\;in\;filtrate}{Mass\;of\;peptide\;in\;initial\;formulation}\times 100\%$$

Study of peptide release from nanoparticles were conducted in 0.1 M acetate buffer of pH 5.5, 0.9% *w*/*v* NaCl. The concentrated NPs were added to the release buffer. At each time point, the particles were separated from the buffer by centrifugal filters. The concentration of released peptide in buffer was calculated using a standard curve on HPLC. 

### 2.7. Cell Uptake Study

To visualise the cell uptake of nanoparticles using fluorescence microscope, rhodamine-PEG-b-PCL was used. Rho-PEG-b-PCL at 10% of the total polymer loading of nanoparticles was used. Specifically, 0.5 mg and 0.125 mg of Rho-PEG-b-PCL was used for DE NPs and POPG NPs, respectively. Human embryonic kidney 293 (HEK293) cells were cultured in 24-well plates overnight. The medium was removed from the wells and washed with 1× phosphate-buffered saline (PBS). Nanoparticles were concentrated using centrifugal filters then mixed with Dulbecco’s Modified Eagle’s Medium (DMEM) at 10% *v*/*v*. After 4 h of incubation, the cells were fixed with 4% PFA and imaged using Leica STED TCS SP5 II Confocal Laser Scanning Microscope (Leica Camera, Wetzlar, Germany). 

### 2.8. Lactate Dehydrogenase (LDH) Cytotoxicity Assay

HEK293 cells were seeded on a 24-well plate at a density of 1 × 10^5^, and 1 µg of *pcDNA3.1-MJD_CAG78_* DNA construct was used to transfect the cells [3]. Different concentrations of P3V8 and QBP1 peptide combination were encapsulated into the following nanoparticles: mPEG-b-PCL (2k 12k) DE NPs, mPEG-b-PCL (2000–12,000) POPG NPs prepared at pH 5, mix ligands PEG-b-PCL (2000–12,000) DE NPs, and mix ligands PEG-b-PCL (2000–12,000) POPG NPs prepared at pH 5. Lactate dehydrogenase (LDH) enzyme activity in the cell culture medium was measured 72 h post plasmid transfection and/or drug treatment using the Cytotox 96 non-radioactive cytotoxicity assay (Promega). The experimental data were all normalised to ‘no transfection’ control.

## 3. Results

### 3.1. Parameters of Double Emulsion (DE) Method Were Optimised to Achieve Suitable Size and Low Polydispersity Index (PDI)

P3V8 and QBP1 are hydrophilic peptides, with solubility in water exceeding 100 mg/mL. The double-emulsion (DE) method can encapsulate hydrophilic drugs, with the matrix composed of hydrophobic polymers. The use of amphiphilic block copolymers further stabilise the colloidal dispersions without the addition of surfactants. As depicted in Figure 1, DE involves three phases into which peptide drugs and polymers are dissolved and sonicated to create nanoparticles. Different amphiphilic polymers were tested to identify the suitable molecular weight and composition that can make nanoparticles of optimal size. mPEG-b-PCL (2000–12,000) had diameter closest to our target of 100–200 nm and lowest polydispersity index (PDI) (Table 1). Another parameter of the DE process is the organic solvent, in which polymer was dissolved. The miscibility of organic solvent and water affects the drug partitioning into different phases and can affect the particle size. Experiments showed that the volume ratio of w1:o:w2 as well as the order of phase addition affects the drug encapsulation efficiency (E.E.). Addition of w1 phase to o phase, then addition of 1st emulsion into w2 phase yielded greatest E.E. Adding o phase to w1 then adding w2 phase into the 1st emulsion yielded lowest E.E. (data not shown). After optimisation, we used volume ratio w1:o:w2 of 0.2:1:4 and added the dispersed phase into the continuous phase to achieve the highest E.E. We tested various organic solvents with mPEG-b-PCL (2000–12,000) and chose dichloromethane based on the diameter and polydispersity of nanoparticles (Table 2). We tested the effects of polymer loading on nanoparticle size but did not observe any obvious trend (Table 3). 

### 3.2. Functionalisation of Polymers Affects Nanoparticle Size, PDI, Drug Encapsulation Efficiency (E.E.), and Drug Release Profile

To enhance cellular uptake, polymers were functionalised with a group of ligands. Denoted ‘mix ligands’ in this paper, a cocktail of targeting ligands was used to assist cellular uptake, as well as BBB crossing and cerebellum targeting for future studies. Mix ligands consisted of 1:1:1:1 mass ratio of TGN-, COG133-, RXR-, and mPEG-b-PCL. These peptides are designed for rendering the nanoparticles to target the brain and enter the cells (although the current report will only evaluate the latter function). COG133 as well as TGN, which also showed cerebellum-targeting capacity, have been demonstrated to cross the blood-brain barrier (BBB) [17,18]. RXR was shown to have significant cell penetration in cerebellum and Purkinje cells [19].

Encapsulating peptide drugs and using ligand-modified PEG-b-PCL increased the particle size and polydispersity (Table 4) (Figure 3). Drug-encapsulated nanoparticles made with mPEG-b-PCL were about 116 nm in diameter and had polydispersity of around 0.13; those made with mix ligands PEG-b-PCL were about 200 nm in diameter and had PDI of around 0.23. Both polymer formulations made nanoparticles of neutral to mildly positive zeta potential. Drug encapsulation efficiency (E.E.) had substantial increases for both peptides when mix ligands PEG-b-PCL was used, compared to when mPEG-b-PCL was used. mPEG-b-PCL nanoparticles had E.E. of around 23% for P3V8 and around 34% for QBP1; mix ligands PEG-b-PCL nanoparticles had E.E. of around 34% for P3V8 and around 46% for QBP1.

Peptide release from mPEG-b-PCL nanoparticles had different trends compared to that from mix ligands PEG-b-PCL nanoparticles (Figure 4). At 6-h time point, P3V8 released from mPEG-b-PCL was around 2% of total encapsulated peptide inside the nanoparticle and started to even out at 24 hat around 3.5%. As for QBP1, around 16.5% of encapsulated peptides were released at 6-h time point and started to even out at 24 h at around 19% release. The release rate and extent for P3V8 and QBP1 from mix ligands PEG-b-PCL DE nanoparticles were not as divergent. P3V8 was released around 4.5% at 6-h time point, and up to around 11% at 48 h. QBP1 was released around 5.8% at 6 h and around 14.7% at 48 h. For both polymer formulations, however, the peptide release did not reach 100% within 48 h; mPEG-b-PCL particles started to even out at 24 h, suggesting that further peptide release is unlikely at later time points. Overall, DE nanoparticles had suitable size, zeta potential, and PDI, but had shortcomings in terms of E.E. and release profile.

### 3.3. Incorporation of Negatively-Charged POPG Lipid Increased Drug E.E. to Almost 100%

To increase E.E. and the extent of drug release, we developed another method using POPG lipid-assisted nanoprecipitation. POPG is a lipid molecule having one negative charge, which can interact with the positive charges of peptides. By adjusting the pH of the aqueous phase in which peptides are dissolved, number of positive charges on P3V8 and QBP1 can be altered; these charges can form ionic bonds with POPG to promote higher E.E. This method promotes drug encapsulation by ionic interaction between peptides and POPG and hydrophobic interaction between POPG and hydrophobic block of polymers, whereas DE promotes drug encapsulation via solvent diffusion. 

Three different pH (pH 3, 5, 7) in the water phase were tested to optimise the nanoparticle size, PDI, zeta potential, E.E. and drug release profile. No obvious trends in PDI and zeta potential were observed among different pH within each polymer type (Table 5) (Figure 5). As for particle size, nanoparticles synthesised in higher pH were greater in diameter for both polymer types. mPEG-b-PCL nanoparticles synthesised in pH 3 had diameter of around 245 nm; in pH 5, around 270 nm; in pH 7, around 320 nm. Mix ligands PEG-b-PCL nanoparticles synthesised in pH 3 had diameter of around 246 nm; in pH 5, around 280 nm; in pH 7, around 320 nm. Drug E.E. trended inversely with the pH; nanoparticles synthesised in higher pH had lower E.E. mPEG-b-PCL nanoparticles synthesised in pH 3 had E.E. of 100% for P3V8 and around 98% for QBP1; in pH 5, around 95% for P3V8 and around 96% for QBP1; in pH 7, around 94% for P3V8 and around 88% for QBP1. Mix ligands PEG-b-PCL nanoparticles synthesised in pH 3 had E.E. of 100% for P3V8 and around 99% for QBP1; in pH 5, around 98% for P3V8 and 100% for QBP1; in pH 7, around 95% for P3V8 and around 85% for QBP1. As for the stability of POPG nanoparticles, among the three pH formulations, particles synthesised in pH 5 were most stable and nanoparticles synthesised in pH 3 were the least stable, disintegrating and creating a viscous dispersion. The stability of particles synthesised in pH 7 was in between that of particles synthesised in pH 3 and pH 5. 

Nanoparticles made in higher pH released P3V8 to a greater extent for both polymer types (Figure 6). Nanoparticles made in pH 7 had release of around 40% at 48 h for mPEG-b-PCL nanoparticles and around 38% for mix ligands PEG-b-PCL nanoparticles. For those made in pH 5 had release of around 31% for mPEG-b-PCL nanoparticles and around 37% for mix ligands PEG-b-PCL nanoparticles at 48 h. Nanoparticles made in pH 3 had the lowest extent of release at 48 h, with mPEG-b-PCL nanoparticles releasing around 28% and mix ligands PEG-b-PCL nanoparticles releasing around 24%. For P3V8, release from POPG nanoparticles started to reach plateau after 48 h, except for mix ligands PEG-b-PCL nanoparticles synthesized in pH 5 and pH 7. From the trend, these two formulations may release more peptides at later time points.

The trend of release from POPG nanoparticles of QBP1 was different from that of P3V8 (Figure 7). For mPEG-b-PCL POPG nanoparticles, those formulated in pH 3 had the greatest extent of QBP1 release at 48-h at around 20%. Particles formulated in pH 5 released the least at around 14%; those synthesised in pH 7 released around 19%. The pH did not affect QBP1 release for mix ligand PEG-b-PCL nanoparticles. At 48 h, nanoparticles synthesised in pH 3 and pH 5 both released around 15% and those synthesised in pH 7 released around 16%. Aside from mix ligands PEG-b-PCL nanoparticles synthesized in pH 3, other formulations did not reach plateau of QBP1 release after 48 hours. From the trend, these formulations are likely to exhibit greater QBP1 release at later time points.

### 3.4. Functionalised Polymer Increased Cell Penetration of DE and POPG Nanoparticles

As cellular uptake of both peptides is required for their bioactivity, we conducted cell uptake studies of our nanoparticles. Both POPG and DE nanoparticles achieved cell penetration when functionalised with mix ligands (Figure 8). Minimal cell uptake occurred with mPEG-b-PCL nanoparticles made by either DE or POPG methods. POPG nanoparticles made at pH 5 had slightly more prominent uptake, but no obvious difference was observed with varying pH or particle synthesis method. 

### 3.5. Encapsulation of Drugs into Either Functionalised DE or POPG NPs Enabled Cell Rescue from CAG RNA Toxicity

Cells transfected with *MJD_CAG78_* DNA construct exhibit both expanded CAG RNA toxicity and polyglutamine protein toxicity [20]. A LDH assay has been applied for evaluating peptide efficacy in inhibiting cell death induced by *MJD_CAG78_* in HEK293 cells [3]. For bioactivity assay, the ‘no transfection’ group was used as negative control; the ‘no treatment’ group was transfected with *MJD_CAG78_* DNA construct but did not receive any treatment (Figure 9). For this bioactivity assay, mix ligand PEG-b-PCL nanoparticles made by DE and POPG at pH5 were tested. As POPG nanoparticles made at pH 5 had the highest stability, this formulation was chosen. Cells treated with DE nanoparticles at 1 μM and 100 nM peptide concentration or POPG nanoparticles at 1 μM had significantly less cell death compared to the control group. Both nanoparticle formulation suppressed cell death while free peptide solutions at the same concentration failed to do so.

## 4. Discussion

The difference in release rate and extent of QBP1 compared to P3V8 in mPEG-b-PCL DE nanoparticles may be due to different partitioning rate of the peptides (Figure 4). QBP1 is more soluble in water than P3V8, thus when nanoparticle is exposed to aqueous buffer, QBP1 is more readily partitioned out from the nanoparticles due to greater solubility. The similar release rate and extent of P3V8 and QBP1 from mix ligands PEG-b-PCL DE may be due to the positive charges of the mix ligands. The positive charges of the mix ligands could have repelled the positively-charged peptides into the core of the polymer matrix; at the core, the effect of high solubility of QBP1 on faster release would be mitigated. A possible cause of incomplete peptide release from both particle formulations is the peptide status after double emulsion process. In an attempt to understand the effect of DE process on peptides, nanoparticles were lyophilised and re-dissolved in organic solvent to remove polymers and precipitate peptides out. When the aqueous phase was tested with HPLC, no observable peaks were present, indicating that the precipitated peptides could not be reconstituted back in water (data not shown). When free peptide powder was mixed with organic solvent and precipitated, the peptides could be completely redissolved in water (data not shown). The peptides released from nanoparticles were also tested with HPLC to verify that the elution time and peak shape were the same as those of peptides that were never encapsulated before. We suspect that the process of DE makes peptides precipitate into less soluble forms, slowing the release and even resulting in incomplete release. Another possible explanation for incomplete release is the high stability of DE nanoparticles. Polycaprolactone has low degradation rate [21,22]. Furthermore, DE nanoparticles had very high stability in water; after weeks of storage in water, no obvious aggregation of particles was observed. Thus, peptide release from nanoparticles mainly rely on the slow diffusion through polymers.

Compared to DE nanoparticles, POPG nanoparticles had overall larger diameter, higher PDI and more negative zeta potential (Table 5). Larger diameter may be due to the bulky presence of POPG inside the particle formulation which could also contribute to lower stability and thus higher PDI of particles. TEM images (Figure 5) suggests that the large diameter also could be due to aggregation of some nanoparticles. The more negative surface charge is likely due to the negative charges of excess POPG. POPG nanoparticles had higher E.E. compared to DE nanoparticles. For both synthesis methods, the E.E. of both peptides within each formulation were similar. The trend in release of P3V8 in relation with pH of w1 phase in POPG nanoparticles (Figure 6) is due to the stability of peptide-POPG interactions inside the nanoparticles. At higher pH, peptides become more neutral, weakening the ionic interaction with POPG. Thus, peptides made in pH 7 will be more readily released compared to those made in pH 3. Compared to DE, POPG nanoparticles release more P3V8 in terms of absolute amount. The lack of relationship between pH and QBP1 release from POPG nanoparticles (Figure 7) can be explained by the different charges of P3V8 and QBP1. While P3V8 has three positive charges at pH 3, QBP1 has one positive charges. At pH 5 and 7, P3V8 has one positive charge, while QBP1 has zero net charge. Thus, for QBP1, encapsulation may be a combination of ionic interaction with POPG and entrapment into the polymer matrix of the particles. This difference in drug encapsulation method can also explain why the extent of release is lower for QBP1 compared to P3V8. As in the case of DE, entrapment in the polymer matrix requires release via diffusion or dissociation of the polymer matrix, both of which can be timely. But compared to DE, POPG nanoparticles release QBP1 more in terms of absolute amount because the polymer matrix is less stable due to incorporation of POPG and gets disrupted faster. 

Both peptides are designed to work intracellularly—P3V8 in the nuclei and QBP1 in the cytoplasm—so cellular uptake is required for bioactivity. The HEK293 cell line was chosen for cell uptake studies to ensure consistency with the bioactivity assay measuring the reduction of cell death after drug treatment (Figure 9). Rhodamine was conjugated to the nanoparticles so that they can be visualised by the red colour if internalized into cells (Figure 8).

Higher potency of DE nanoparticles is suggested by the results as lower concentration of peptide-encapsulated DE nanoparticles were needed to significantly suppress cell death compared to POPG nanoparticles (Figure 9). Taken with the release data of Figure 6 and Figure 7, our results demonstrate that encapsulation of peptides into nanoparticles preserves the bioactivity of the peptide drugs in alleviating polyglutamine-induced toxicity, and that the nanoparticles achieve significant intracellular release of potent peptides into cells.

Although the peptide drugs were designed to ultimately target cerebellum, HEK293 cells were appropriate for our cell uptake study and bioactivity assay for the following reason. The mixture of ligands consists of TGN, COG133, and RXR. TGN has been reported to assist BBB crossing and cerebellum targeting by utilising clathrin- and caveolae-mediated endocytosis [23]. As clathrin is a ubiquitous protein [24] and caveolae is widely expressed in various tissues [25], the use of HEK293 cells for our studies could assess the potential of the nanoparticles for intracellular delivery. The method of cell penetration of RXR is yet to be fully understood, but speculated to be similar to that of other arginine-rich cell penetrating peptides [19]. Ability for cell uptake by nanoparticles presenting mix ligands enable the internalization of peptide drugs and is therefore essential for their functions. We did not intend to evaluate the cell penetration efficacy nor BBB crossing potential of individual ligands in this paper, but to validate the possibility of co-encapsulating two hydrophilic peptide drugs into polymeric nanoparticles; to characterize the particle properties and release profile; and to evaluate cell uptake and therapeutic effect of nanoparticles.

The higher therapeutic potential of DE nanoparticles despite similar cellular penetration compared to POPG nanoparticles may be attributed to more neutral zeta potential. Nanoparticles with more negative zeta potential demonstrated less endosomal escape than those with more positive zeta potential [26,27]. As POPG nanoparticles have more negative zeta potential compared to DE nanoparticles, they may have escaped endosomes at lower efficiency, resulting in lower concentration of peptides in cells to rescue them from toxicity. Another possibility is that POPG nanoparticles disintegrated and released the peptide drugs too quickly due to its lower stability, disenabling effective cell uptake of the peptide drugs. Also, as member of a major cell membrane components, glycerophospholipids, POPG may dissociate from the particle and exchange with those in the cell membrane, accelerating the POPG nanoparticle disintegration [28,29]. The cell uptake study was conducted over 4 h while the bioactivity assay was conducted over 72 h. Cell uptake of the stable, peptide-encapsulated DE nanoparticles may be more prominent compared to POPG nanoparticles at 72 h unlike at 4 h.

## 5. Conclusions

In this paper, we present two methods of encapsulating two hydrophilic peptide drugs for co-delivery using polymeric nanocarriers. Both methods can simultaneously deliver different peptide drugs to meet the need of combination therapy, and for the first time, with relevance for treatment of neurodegenerative polyglutamine diseases. Nanoparticles made by the two methods reported have distinctive characteristics. Nanoparticles prepared by double emulsion are more uniform in size distribution, more stable, and exhibit higher bioactivity. Nanoprecipitation in the presence of POPG produced nanoparticles with higher encapsulation efficiency and faster drug release. 

## Figures and Tables

**Figure 1 polymers-11-00288-f001:**
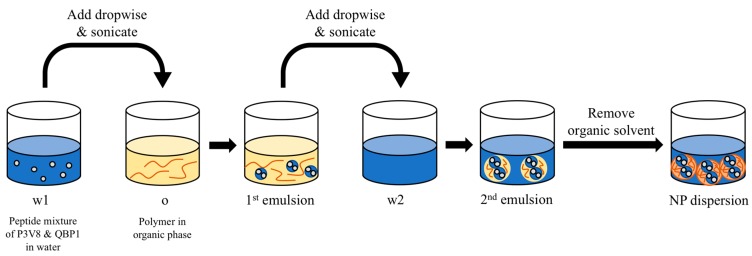
Preparation of DE NPs by the double-emulsion method.

**Figure 2 polymers-11-00288-f002:**
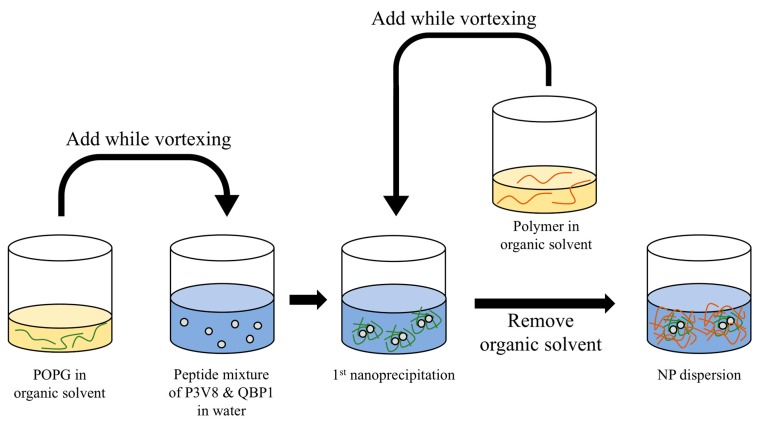
Two-step nanoprecipitation method with POPG addition for the preparation of POPG NPs.

**Figure 3 polymers-11-00288-f003:**
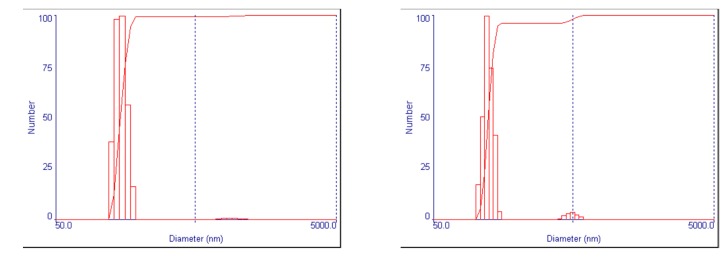
Characterization of DE NP size and morphology via dynamic light scattering (DLS) and transmission electron microscopy (TEM). DLS profiles of DE NPs show uniformity of nanoparticle size. Nanoparticles diameter as measured by TEM is similar to the mean diameter measured by DLS.

**Figure 4 polymers-11-00288-f004:**
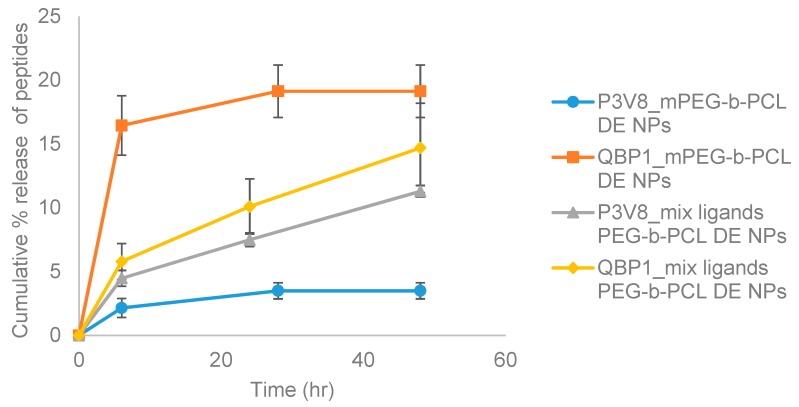
Mean cumulative % release of peptides from co-encapsulated DE NPs of various formulations. mPEG-b-PCL DE NPs refers to nanoparticles made with mPEG-b-PCL by DE method. Mix ligands PEG-b-PCL DE NPs refers to nanoparticles made with mix ligands conjugated PEG-b-PCL using DE method. (*n* = 3).

**Figure 5 polymers-11-00288-f005:**
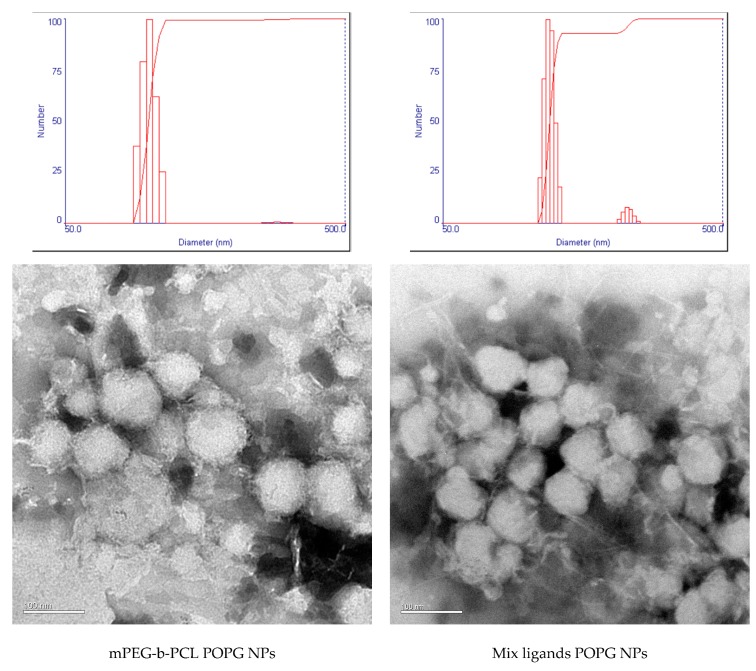
Characterization of POPG NP size and morphology via DLS and TEM. TEM images of nanoparticles reveal that the large diameters measured by DLS may be due to aggregation of nanoparticles.

**Figure 6 polymers-11-00288-f006:**
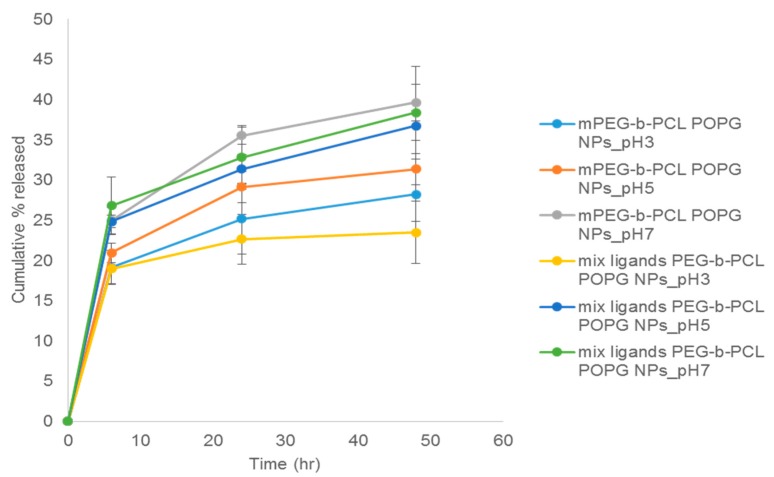
Mean cumulative release of P3V8 from co-encapsulated POPG NPs of various formulations. mPEG-b-PCL POPG NPs refers to nanoparticles made with mPEG-b-PCL using nanoprecipitation method with POPG. Mix ligands PEG-b-PCL POPG NPs refers to nanoparticles made with mix ligands PEG-b-PCL using nanoprecipitation method with POPG. The following pH value refers to pH of aqueous phase used during nanoprecipitation (*n* = 3).

**Figure 7 polymers-11-00288-f007:**
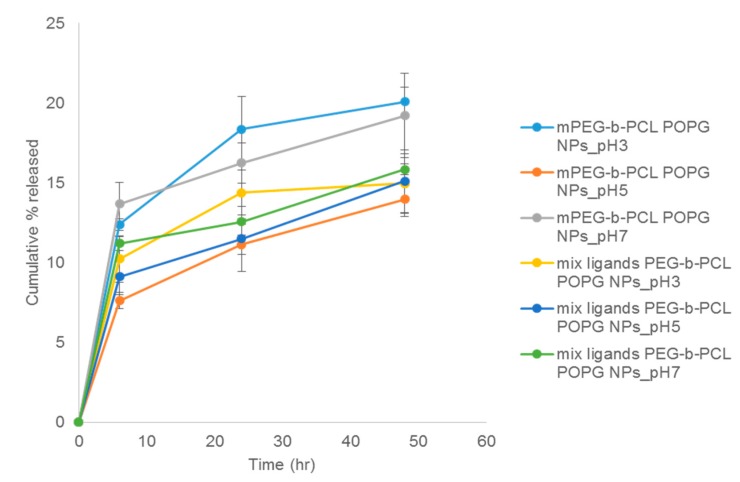
Mean cumulative release of QBP1 from co-encapsulated POPG NPs of various formulations. mPEG-b-PCL POPG NPs refers to nanoparticles made with mPEG-b-PCL using nanoprecipitation method with POPG. Mix ligands PEG-b-PCL POPG NPs refers to nanoparticles made with mix ligands PEG-b-PCL using nanoprecipitation method with POPG. The following pH value refers to pH of aqueous phase used during nanoprecipitation (*n* = 3).

**Figure 8 polymers-11-00288-f008:**
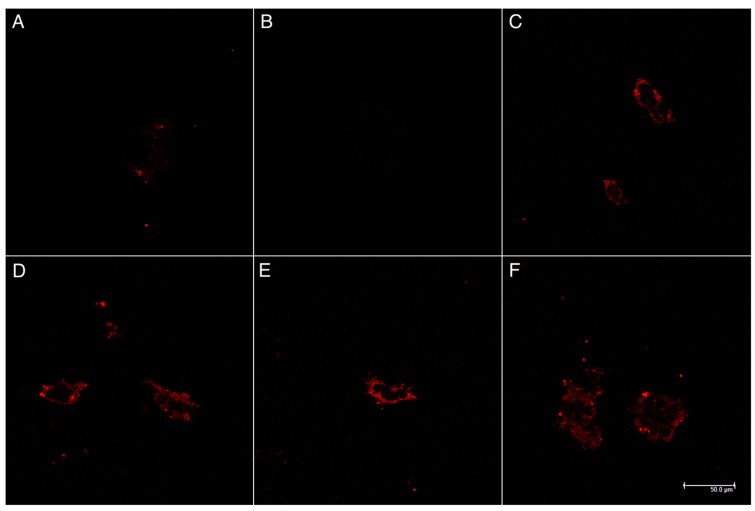
Stimulated emission depletion microscopy (STED) images of HEK293 incubated 4 h with NP treatment. Red fluorescence from Rhodamine-PEG-b-PCL to localize nanoparticles. (**A**) mPEG-b-PCL DE NPs; (**B**) mPEG-b-PCL POPG NPs; (**C**) mix ligands PEG-b-PCL DE NPs; (**D**) mix ligands PEG-b-PCL POPG NPs, pH 3; (**E**) mix ligands PEG-b-PCL POPG NPs, pH 5; and (**F**) mix ligands PEG-b-PCL POPG NPs, pH 7.

**Figure 9 polymers-11-00288-f009:**
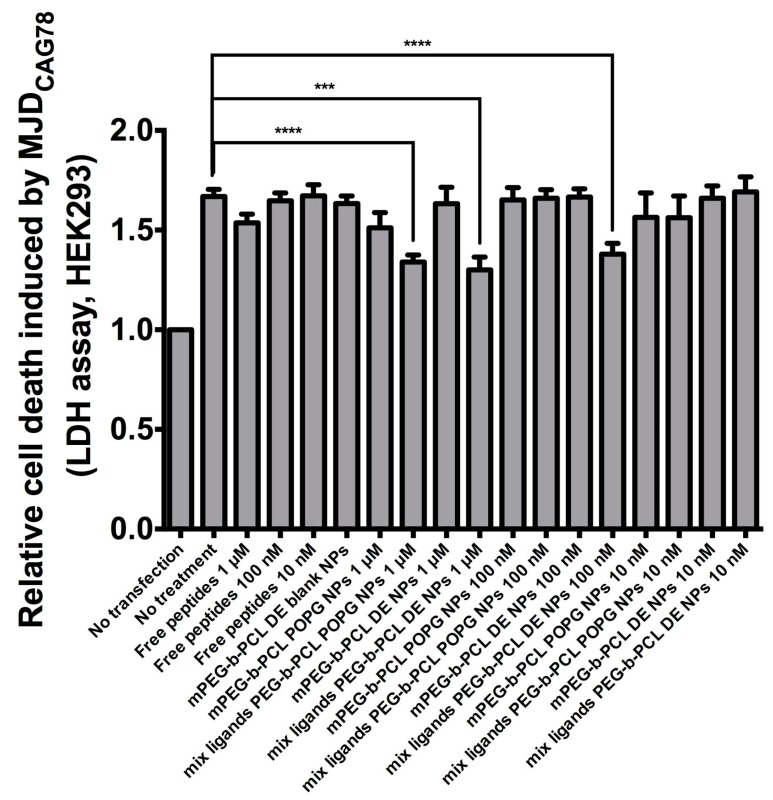
Effect of peptide-encapsulated NPs on suppressing cell death in MJD_CAG78_-expressing HEK293 cells. Free peptides refers to solution of P3V8 and QBP1 (1:1 mass ratio). Blank NPs refers to nanoparticles without peptide encapsulation. All other NPs are encapsulated with mixture of P3V8 and QBP1 (1:1 mass ratio). Concentration refers to final peptide concentration applied to cells (*n* = 3). mean ± SD. *** *p* < 0.0005, **** *p* < 0.0001.

**Table 1 polymers-11-00288-t001:** Characterization of blank DE nanoparticles.

Polymer	Mean Diameter by Number (nm)	Mean Diameter by Intensity (nm)	Mean PDI
mPEG-b-PCL (2000–12,000)	58.8 ± 34.4	139.2 ± 0.7	0.16 ± 0.02
mPEG-b-PLGA (5000–45,000)	77.8 ± 14.8	640.3 ± 24.1	0.21 ± 0.04
mPEG-b-PLGA (2000–11,500)	235.9 ± 59.1	281.9 ± 7.8	0.20 ± 0.04
mPEG-b-PLLA (5000–20,000)	129.6 ± 52.0	316.5 ± 5.7	0.25 ± 0.06

**Table 2 polymers-11-00288-t002:** Characterization of blank DE nanoparticles synthesized with various organic solvents.

Polymer	Organic Solvent	Mean Diameter by Number (nm)	Mean Diameter by Intensity (nm)	Mean PDI
mPEG-b-PCL (2000–12,000)	Dichloromethane	58.8 ± 34.4	139.2 ± 0.7	0.16 ± 0.02
	Chloroform	209.0 ± 109.4	173.3 ± 0.8	0.23 ± 0.02
	Ethyl acetate	256.4 ± 106.7	228.1 ± 11.4	0.31 ± 0.02
	DCM+acetone (1:1, *v*/*v*)	87.7 ± 1.8	299.8 ± 42.1	0.38 ± 0.00
	Tetrahydrofuran	76.3 ± 29.6	179.4 ± 17.3	0.18 ± 0.02
	Acetonitrile	72.1 ± 4.2	136.8 ± 7.2	0.28 ± 0.01

**Table 3 polymers-11-00288-t003:** Characterization of blank DE nanoparticles synthesized with various polymer loading.

Polymer	Polymer Loading (mg/mL)	Mean Diameter by Number (nm)	Mean Diameter by Intensity (nm)	Mean PDI
mPEG-b-PCL (2000–12,000)	1	69.3 ± 22.1	146.0 ± 2.7	0.20 ± 0.01
	5	92.2 ± 3.5	132.0 ± 0.3	0.21 ± 0.00
	10	70.7 ± 13.8	128.3 ± 1.2	0.21 ± 0.02

**Table 4 polymers-11-00288-t004:** Characterization of PEG-b-PCL (2000–12,000) nanoparticles made by double emulsion (DE NPs) (Mean ± SD) (*n* = 6).

Polymer	Mean Diameter by Number (nm)	Mean Diameter by Intensity (nm)	Mean PDI	Mean Zeta Potential (mV)	Mean E.E. (%)	
mPEG-b-PCL	59.0 ± 53.8	116.7 ± 10.1	0.13 ± 0.07	−0.09 ± 0.49	P3V8	22.90 ± 8.91
					QBP1	34.20 ± 3.70
Mix ligands PEG-b-PCL	53.1 ± 16.8	201.3 ± 1.5	0.23 ± 0.02	0.02 ± 0.05	P3V8	34.04 ± 4.88
					QBP1	46.26 ± 4.50

**Table 5 polymers-11-00288-t005:** Characterization of PEG-b-PCL (2000–12,000) nanoparticles made by nanoprecipitation with lipid POPG (POPG NPs). (Mean ± SD) (*n* = 3).

Polymer	pH of Water Phase	Mean Diameter by Number (nm)	Mean Diameter by Intensity (nm)	Mean PDI	Mean Zeta Potential (mV)	Mean E.E. (%)	
mPEG-b-PCL	3	63.6 ± 17.2	245.2 ± 12.4	0.34 ± 0.01	−52.1 ± 2.9	P3V8	100.00 ± 0.00
						QBP1	98.06 ± 2.37
	5	57.6 ± 20.5	269.9 ± 21.5	0.34 ± 0.02	−44.1 ± 4.6	P3V8	94.85 ± 0.41
						QBP1	96.07 ± 0.26
	7	87.3 ± 26.5	319.0 ± 60.8	0.33 ± 0.03	−44.8 ± 7.9	P3V8	94.33 ± 0.11
						QBP1	88.48 ± 1.97
Mix ligands PEG-b-PCL	3	80.8 ± 16.5	246.2 ± 14.2	0.31 ± 0.01	−38.7 ± 0.8	P3V8	100.00 ± 0.00
						QBP1	99.36 ± 0.56
	5	74.9 ± 37.3	284.1 ± 40.3	0.31 ± 0.03	−47.6 ± 4.4	P3V8	98.27 ± 0.33
						QBP1	100.00 ± 0.00
	7	63.5 ± 1.9	321.0 ± 107.3	0.32 ± 0.05	−42.0 ± 2.4	P3V8	94.68 ± 0.69
						QBP1	84.89 ± 5.57

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
