# Peer review of "Co-Encapsulation and Co-Delivery of Peptide Drugs via Polymeric Nanoparticles"

_polymers, 2019, doi:10.3390/polym11020288_

Round 1
Reviewer 1 Report
1) in the introduction: "Nanoparticles can be synthesised to co-encapsulate P3 and QBP1 into each nanoparticle"- I think it should be said c-encapsulation into the same nanoparticle.
2) The rationale behind corporation - should be the rationale behind incorporation.
3) The results section is well written but it is like one single section. It will be hard for the audience to follow the findings. I suggest the authors should subdivide it with side headings that provide a takehome message from that paragraph.
4) The authors should perform TEM and demonstrate the size and shape of the nanoparticles.
5) Please provide the representative DLS profiles for each and every nanoparticle.
6) The authors discussed about precipitated and unreleasable proteins in DE particles, I think if the authors provide more information by investigating what changes has been happened to the proteins, if there any conformational changes or andy chemical changes, if they can use NMR and HPLC and compare it with the original peptide profiles will increase the potential of the current work.
Author Response
Thank you for your feedback. The replies to your feedback has been uploaded as a word document.

Reviewer 2 Report
In this work two peptide therapeutics, P3 and QBP1 have been microencapsulated by double emulsion-solvent evaporation and nanoprecipitation methods. The manuscript overall is well-written and contains sufficient information on the preparation, and the theories and explanations are generally clear. However, in some points it should be clarified and some more experiments are suggested.
Major remarks:
1. Page 3: nanoprecipitation method is not reproducible. How much peptide was dissolved in the water phase? What was the ratio of water and oil phase? It would be clearer to give more experimental data similar to the above for double emulsion method as well in the experimental section, which can be found only in the result section.
2. Page 3, line 110: ’Experiments were done to verify that peptides do not bind to the centrifugal filters and that nanoparticles can be effectively retrieved from filters.’ These experiments should be described.
3. Page 3, line 120: ’Rho-PEG-b-PCL at 10% of the NP polymer loading was used.’ It is not clear what the authors mean under ’NP polymer loading’ in the cellular uptake studies.
4. How was the size of the NPs measured? What do ’mean diameter’ and ’mean effective diameter’ mean in Tables 1-4? Mean diameter by intensity, number or volume?
5. Page 5, from line 166: ’ Encapsulating peptide drugs and using ligand-modified PEG-b-PCL did not largely affect the particle size or polydispersity (Table 4). Drug-encapsulated nanoparticles made with mPEG-b-PCL were about 116 nm in diameter and had polydispersity of around 0.13; those made with mix ligands PEG-b-PCL were about 200 nm in diameter and had PDI of around 0.23.’ Average size change from 116 nm to 200 nm is a significant change of size. Similarly in the next sentence, drug encapsulation efficiency change from 23 % to 34 % and 46 % is not a slight but a substantial enhancement.
6. Scanning or transmission electron microscopic images should be provided for the nanoparticles prepared by the two different methods for a chosen nanoparticle composition without and with ’mix ligands’.
Minor comments:
1. Page 1, line 27: ’have rose’ must be corrected to ’have risen’.
2. Page 1, line 29: ’two peptides drugs’ should be changed to ’two peptide drugs’.
3. Page 1, line 30: ’have showed’ should be substituted by ’have shown’.
4. Page 2, line 84: ’2.2. Synthesis of mPEG-b-PCL, allyl-PEG-b-PCL, ligand-modified PEG-b-PCL’. It must be described also here which ligands were conjugated to the PEG-b-PCL.
5. In Table 3 instead of polymer loading in mg polymer concentration of organic phase should be given in mg/ml.
6. Page 9, line 268: ’…while free peptide solutions at the same concentration failed to.’ must be corrected to e.g. ’…failed to do it.’
7. Page 11, line 327: ’for the following reasoning’ must be corrected to ’for the following reason’.
Author Response
Thank you for your feedback. Replies to your feedback has been uploaded as a Word document.

Round 2
Reviewer 1 Report
The authors have addressed most of the concerns of the reviewers.
Below are few changes recommended before accepting the current form of the manuscript.
1) Please include the TEM images and a brief writeup discussion in the currently updated manuscript.
2) The same goes for the DLS profiles.
3) The authors have performed HPLC analysis of the un rproteinle protien to investigate if there is any change. Please include the discussion lines from the author's response to the manuscript. if possible include the HPLC profiles.
Author Response
1) Please include the TEM images and a brief writeup discussion in the currently updated manuscript.
Thank you for your comment. The TEM images and a brief writeup discussion have been added to the manuscript.
2) The same goes for the DLS profiles.
Thank you for your comment. The DLS profiles and a brief writeup discussion have been added to the manuscript.
3) The authors have performed HPLC analysis of the un rproteinle protien to investigate if there is any change. Please include the discussion lines from the author's response to the manuscript. if possible include the HPLC profiles.
Thank you for your comment. Discussion on the HPLC analysis of released peptides has been added to the manuscript.

Reviewer 2 Report
The corrections and answers are fine. Two little corrections are still necessary. Dynamic light scattering method and the used instrument must be added to the methods. In Table 1-5, 'mean diameter' should be changed to 'mean diameter by number', and 'mean effective diameter' should be changed to 'mean diameter by intensity'.
Author Response
The corrections and answers are fine. Two little corrections are still necessary. Dynamic light scattering method and the used instrument must be added to the methods. In Table 1-5, 'mean diameter' should be changed to 'mean diameter by number', and 'mean effective diameter' should be changed to 'mean diameter by intensity'.
Thank you for your comments. The DLS method and the used instrument has been added to the method section and the suggested changes for Table 1-5 have been made.
